# LatticeGen: A Cooperative Framework which Hides Generated Text in a Lattice for Privacy-Aware Generation on Cloud

## Abstract

In the current user-server interaction paradigm of prompted generation with large language models (LLM) on cloud, the server fully controls the generation process, which leaves zero options for users who want to keep the generated text to themselves. We propose LatticeGen, a cooperative framework in which the server still handles most of the computation while the user controls the sampling operation. The key idea is that the true generated sequence is mixed with noise tokens by the user and hidden in a noised lattice. Considering potential attacks from a hypothetically malicious server and how the user can defend against it, we propose the repeated beam-search attack and the mixing noise scheme. In our experiments we apply LatticeGen to protect both prompt and generation. It is shown that while the noised lattice degrades generation quality, LatticeGen successfully protects the true generation to a remarkable degree under strong attacks (more than 50% of the semantic remains hidden as measured by BERTScore).

## 1 Introduction

Many of the high-performing large language models (LLMs) need to be deployed on cloud servers, whether they are open-sourced but have an intensive need for computation (Zhao et al., 2023; Kaplan et al., 2020), or behind a paywall like ChatGPT (OpenAI, 2023). This raises new privacy challenges (Li et al., 2021; Yu et al., 2021; Kerrigan et al., 2020), since users have to send or receive their data to/from cloud providers.

In this work we focus on a popular interaction paradigm between end users and a server hosting an LLM on cloud named prompted generation: The user sends server a prompt, which is usually an instruction (Chung et al., 2022) or a beginning of a document, and the server, who fully controls the generation process, sends user back the generated text from the LLM. Both the prompt and the generation are raw texts which are completely transparent and accessible to the server, leaving zero options for users who want to keep the generated text to themselves. This is a serious problem as LLMs become widely deployed in professional and social applications. [1]

Existing work in privacy-aware NLP (Qu et al., 2021; McMahan et al., 2017) mostly focuses on protecting user data for training (e.g., federated learning (Huang et al., 2020)) or inference, and the majority of works focus on natural language understanding (NLU) tasks (Feyisetan et al., 2020). We argue that in prompted generation, there are many scenarios in which user prompts *as well as the generated contents* need obfuscation, because they can directly affect the user's decisions (as we discuss in §2).

With the goal of preventing the server from gaining complete knowledge of the generated text and prompt, we propose LatticeGen (Figure 1), a user–server interaction framework in which the user and server conduct prompted generation token-by-token in **a cooperative way**. We summarize our key contributions below:

- The key idea of LatticeGen (§3) is that in each time-step, the user sends the sever not one, but $N$ tokens (thus the name *lattice*), in which one is true and others act as noise. The server

---

[1] See §F for a more involved discussion about the current industry state.

does LLM inference and sends user back the next-token prediction distributions for all $N$ tokens, which are used by user to sample the true and noise tokens for the next time-step.

- Considering potential attacks from a hypothetically malicious server and how the user can defend against it (§4), we propose the repeated beam-search attack and the mixing noise scheme.

- We apply LatticeGen to the task of creative writing (Fan et al., 2017). Our experiments (§5) show that while the noised lattice degrades generation quality, LatticeGen successfully prevents a malicious server from recovering the true generation to a remarkable degree (more than 50% of the semantic remains unknown as measured by BERTScore). [2]

## 2 Motivation and Preliminaries

**Generated Text (also) Needs Obfuscation**   In the current user–server interaction paradigm, the user sends the server a prompt which is usually the beginning of a dialogue, story or instruction, then the server generates a complete response using the process described below, and sends it back to the user. Both the prompt and generation are directly available to the server in raw text format.

This paper contends that in privacy-aware settings, generated texts, as well as user prompts, require a privacy protection mechanism. A key reason is that the generation from the LLM can **affect the user's private decisions**: e.g., a customer is likely to go to the restaurant suggested by the LLM; an engineer could adopt the approach proposed by the LLM; a writer could take inspiration from outputs provided by the LLM. Also see §F for recent privacy-related incidents with ChatGPT or Bard. The obfuscation provided by LatticeGen makes it harder for a hypothetically malicious server to infer the user's actions after interacting with the LLM.

On the other hand, under stochastic sampling with ample diversity in open-ended tasks (Dai et al., 2019; Fan et al., 2017), the generated text is in general unique, and can not be directly replicated even if the prompt is known to the server. In LatticeGen, this is ensured by sampling with a private random seed controlled by the user (§3).

**Standard Autoregressive LM Generation**   We assume the server-side LLM is an autoregressive LM, i.e., it generates tokens one at a time and from left to right. We denote the LLM as $P_M$ with parameter set $\theta$, the vocabulary as $V$, the generated token at time-step $t$ as $w_t$, and the given prompt as $p$. In this work we regard the prompt as part of generation, therefore, $w_t := p_t$ for $1 \leq t \leq \text{len}(p)$. On each time-step $t > \text{len}(p)$, we sample the next token $w_t$ from $P_M(\cdot|w_{0..t-1})$ by calling a sampling algorithm such as top-$k$ (Fan et al., 2017) or nucleus sampling (Holtzman et al., 2020). $w_0$ is the <bos> token.

**The Lattice Structure**   A key concept in our proposed framework is the lattice (Young et al., 2006), which is a graphical structure widely used in structured prediction problems to represent a range of hypotheses. In this work we adopt a simple linear-graph form of lattice which is known as the confusion network (Mangu et al., 1999). For convenience, we will just refer to it as the *lattice*, and an example is shown in the left part of Figure 1. For a width-$N$ lattice (or an $N$-lattice for short), each time-step contains $N$ token options and we denote them as $\{w_t^1, ..., w_t^N\}$ (see Figure 1 for an example with $N = 2$). Therefore, a $N$-lattice with length $T$ (denoted as $W_T^N$) represents $N^T$ possible sequence combinations.

In our proposed LatticeGen protocols (§3.1), for each time-step $t$, the user possesses the "true" generated token denoted as $w_t^1$. And the other $N-1$ tokens $\{w_t^2, ..., w_t^N\}$ are be referred to as "noise" tokens. To prevent the server from knowing which one is the true token, the user will randomly shuffle the list before sending it to the server, denoted as $[\tilde{w}_t^1, ..., \tilde{w}_t^N]$. We will also use the notation $\tilde{W}_t^N$ to emphasize that the tokens in the lattice are shuffled in each time-step.

**LM Finetuning and Inference with the Linearized Lattice Format**   As a prerequisite for LatticeGen, we need the transformer LM (Vaswani et al., 2017) to be able to do inference based on a given lattice and we achieve that by finetuning the base LLM $P_M$ to make next-token prediction with the *linearized lattice format*. Below we first introduce this format, and describe the finetuning objective.

---

[2]Our code and data will be available in the public version of this manuscript.

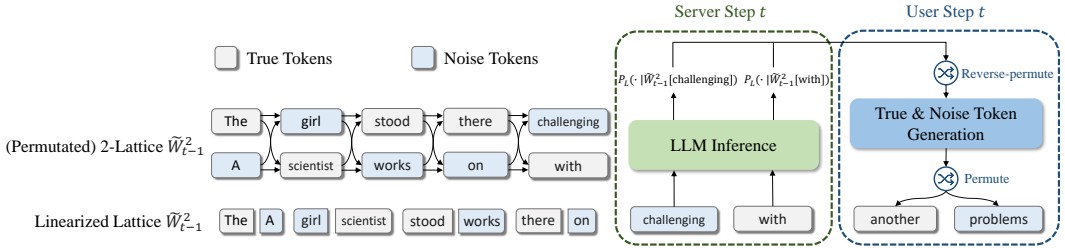

Figure 1: User-Server interaction under LatticeGen for time-step $t$. The server controls the LLM $P_L$ and conducts the inference computation and sends user the next-token prediction distribution for each received token. The user conducts the true and noise token generation, and sends server a randomly permutated list of tokens for the next time-step. The task is creative writing.

As the name suggests, we conduct a simple *linearization* operation before feeding it to the LM, in which the token options on each time-step are linearized and concatenated into a sequence of length $T \times N$ (see Figure 1 for an example):

$$\text{linearize}(\tilde{W}_T^N) = [\texttt{<bos>}] + \text{concat}_{t=1}^T([\tilde{w}_t^1, ..., \tilde{w}_t^N]). \tag{1}$$

To simplify notation, when the lattice appears as part of the history in an LLM inference, we assume it is linearized. The terminology *position* is used to refer to the index of a token in the linearized lattice. We use the notation $P_L(\cdot|\tilde{W}_T^N[\tilde{w}_t^i])$ with $T \geq t$ to refer to the next-token prediction distribution on the position of token $\tilde{w}_t^i \in \tilde{W}_T^N$ outputted by $P_L$. An illustration of this notation is given in Figure 1.

In §A we describe in detail how the LLM can be finetuned to accept a linearized lattice as input, and we denote the lattice-finetuned LLM as $P_L$, which is trained to give the next-token prediction for every token in the linearized lattice. Here we provide a high-level description. For each data sample $w^1$, we construct and linearize a noised lattice by applying a simple synonym noise scheme (also see §3.1). During training, the next-token target for the positions of true data tokens $P_L(\cdot|\tilde{W}_T^N[\tilde{w}_t^1])$ are set to its next true token $w_{t+1}^1$, while all noise tokens do not get training signal (Figure 4, §A). [3]

## 3 LATTICEGEN

To prevent the server from gaining full knowledge of the generation and prompt, LatticeGen makes several core changes to the user–server interaction. **Instead of letting the server handle the whole generation process alone, the user and server will conduct the generation token-by-token in a cooperative way on each time-step**. In particular, the user will *hide* the true generation in a width-$N$ *lattice*, where $N \geq 2$ is a hyperparameter controlling the number of noise tokens.

### 3.1 PROTOCOLS

On a high level, the server who possesses the lattice-finetuned LLM $P_L$ (the training is detailed in §A) still handles most of the computation, while the user controls the key sampling operation and expands the lattice to the next time-step. The server needs to share the vocabulary $V$ with the user, but all other parameters or configurations of the LLM are not shared. We describe the protocols below.

For simplicity, we first ignore the prompt part and assume the generation starts at the first token. In the beginning $t = 0$, the server begins with an empty local lattice, and the user sends a $w_0 = \texttt{<bos>}$ token to the server. We divide the user–server interaction at each time-step $t \geq 1$ into a *server step* and a *user step*, illustrated by Figure 1 (also see Algorithm 1 in §B).

**Server Step** From the last time-step, the server receives $N$ permutated tokens $\{\tilde{w}_{t-1}^1, ..., \tilde{w}_{t-1}^N\}$ (only the first time-step is special where only $\texttt{<bos>}$ is received) and expands its local lattice to $\tilde{W}_{t-1}^N$. The permutation is done by the user from the last time-step, and the server does not know which received token is the true token. The server then computes the respective next-token prediction distribution for all $N$ tokens with the LLM. More concretely, the lattice $\tilde{W}_{t-1}^N$ is linearized and fed to $P_L$, which outputs $\{P_L(\cdot|\tilde{W}_{t-1}^N[\tilde{w}_{t-1}^i])\}_{i=1}^N$.

---

[3] Since the true and noise tokens are shuffled, the LLM learns to predict the next token for every position.

With a properly finetuned LLM, this can be done efficiently with one pass of model inference. We defer the details of finetuning and inference (both conducted by the server) to §A. The server represents the distributions as $N$ length-$|V|$ vectors, and sends them back to the user.

**User Step** Upon receiving the list of distribution vectors from the server, the user applies the reverse permutation mapping (saved from the last time-step) and obtains $\{P_L(\cdot|\tilde{W}_{t-1}^N[w_{t-1}^i])\}_{i=1}^N$. Knowing that $w_{t-1}^1$ is the true token, and the user samples $w_t^1$ from $P_L(\cdot|\tilde{W}_{t-1}^N[w_{t-1}^1])$. The user also generates $N-1$ "noise" tokens $\{w_t^2, ..., w_t^N\}$ with a certain noise scheme.

How to generate noise tokens is a key part of making our framework robust to potential attacks from the server side. For now, we assume a simple synonym noise scheme in which we use synonyms of the true token. Concretely, $w_t^2$ is randomly sampled from $S$ tokens which the closest embedding with $w_t^1$ measured by cosine similarity. In our experiments we set $S = 5$. [4] In practice this simple noise scheme will be vulnerable to attacks from a malicious server. See §4 for discussions on attacks and more advanced noise schemes for defense.

With a private random seed, the user permutates the token list and sends it to the server. [5] We denote the permutated list by $[\tilde{w}_t^1, ..., \tilde{w}_t^N]$. The reverse mapping of the permutation is saved by the user for the next time-step and is not shared with the server. This concludes the user–server interaction in time-step $t$.

**Incorporating Prompts (User)** The incorporation of prompts is quite straightforward by regarding it as a prefix of the generation, and the content in the prompt can also be noised and protected by LatticeGen. See §B.1 for implementation details.

We summarize the LatticeGen protocols as pseudo-code in Algorithm 1 (§B). The discussion on the network communication cost between user and server is deferred to §B.3 to save space.

## 3.2 COMPARISON WITH STANDARD LM: HISTORY NOISED WHILE LOCALLY SHARP

It is helpful to formulate a comparison between LatticeGen ($P_L$) and generation from a standard autoregressive LM $P_M$. For simplicity, we ignore the noise generation (i.e., lattice-building) part, and only care about how the true tokens are generated with $P_L$. Under this simplification, the probability of generating a true sequence $w$ is:

$$\log P_L(w) \approx \sum_{t=1}^T \log P_L(w_t|\tilde{W}_{t-1}^N[w_{t-1}]), \tag{2}$$

where the forming process of $\tilde{W}_{t-1}^N$ (noise tokens and permutation) at each time-step is omitted.

For comparison, the log-probability of generating $w$ with the standard model $P_M$ is:

$$\log P_M(w) = \sum_{t=1}^T \log P_M(w_t|w_{0...t-2}, w_{t-1}). \tag{3}$$

Comparing the above two equations with similar structure, it should be clear that **what LatticeGen does is essentially blurring the token history** $w_{0...t-2}$ **by the noised lattice** $\tilde{W}_{t-2}^N$. Therefore, increasing the number of noise tokens gives better protection for the true token sequence, but at the same time degrades the LM's performance.

**While the history is blurred, the local sharpness (Khandelwal et al., 2018) is preserved by LatticeGen:** From Equation 2, the exact last token $w_{t-1}$ is provided to the model. Therefore, in the worst-case scenario (zero utilization of non-immediate history), LatticeGen is at least as strong as a bigram LM (or a trigram LM when we use bigram units, see §3.3).

## 3.3 INCORPORATING BIGRAM UNITS

In the formulations described above, when the server is doing LLM inference on time-step $t$, only the last token $\tilde{w}_{t-1}^i$ is locally "exact" or "sharp" (explained in §3.2) while all other context tokens are

---

[4]In practice, we exclude the first ten closest token in $V$, as their surface forms are usually very close to the true token, making the obfuscation useless (e.g., only different in capitalization).

[5]The seed can be $t$ multiplied by a large prime number only known to the user.

noised by the lattice. In other words, the inference unit is unigram-level. Naturally, this would lead to serious degradation of generation quality.

To alleviate it, we explore a variant in which we expand the unit from unigram (one token) to bigram (two adjacent tokens). While **the lattice is still one token per time-step**, the user enumerates all $N^2$ potential bigram combinations of $w_{t-2}$ and $w_{t-1}$ and asks the server LLM to return the next-token prediction distribution for each bigram. The formulations for the bigram variant are highly similar to the unigram case and we defer them to §B.2 (also see Figure 6).

## 4  ATTACK AND DEFENSE

In this section, we discuss the potential attack algorithms from a hypothetically malicious server to reverse-engineer the true token sequence $\{w_t^1\}_{t=1}^T$ hidden in the lattice $\tilde{W}_T^N$, and the user's noise generation schemes as defense. We first establish metrics for measuring the strength of attacks.

**Metrics**  Given a lattice $\tilde{W}_T^N$, the attacker's target is to reverse-engineer a hypothesis sequence $\hat{w}$ with $\hat{w}_t \in \{\tilde{w}_t^1, ..., \tilde{w}_t^N\}$ having biggest overlap with the true generation $w^1$. We define a simple *true-ratio* metric to measure the strength of the attack algorithm:

$$\text{true-ratio}(\hat{w}, w^1) = \frac{\sum_{t=1}^T \mathbb{1}_{\hat{w}_t = w_t^1}}{T}. \tag{4}$$

In the repeated beam search attack described below, the result of the attack algorithm is not only one but $N$ sequences $\{\hat{w}^i\}_{i=1}^N$ which spans the whole lattice (i.e., $\{\hat{w}_t^i\}_{i=1}^N = \{\tilde{w}_t^i\}_{i=1}^N$). In this case, we argue that the defending noise scheme should prevent *any* of the hypothesis from having a high overlap with the true sequence, and measure it with the maximum true-ratio: [6]

$$\text{max-true-ratio}(\{\hat{w}\}_{i=1}^N, w^1) = \max_i \frac{\sum_{t=1}^T \mathbb{1}_{\hat{w}_t^i = w_t^1}}{T}. \tag{5}$$

And we state without proof that $\frac{1}{N}$ is a lower bound for max-true-ratio for any noise scheme, which provides an intuition of why larger $N$ would better protect the true sequence.

Albeit intuitive, a big weakness of the true-ratio metric is that it only considers exact matches and does not reflect the semantic similarity between the hypothesis and the true generation. Therefore, in our experiments we will also use an embedding-based metric BERTScore (Zhang* et al., 2020) to measure the leaked information on semantics. Similar to true-ratio, BERTScore is larger than zero and has a maximum value of 1 (we refer readers to its paper for details). We define max-BERTScore in the same fashion as max-true-ratio and we omit the formulation for brevity.

### 4.1  THE REPEATED BEAM-SEARCH ATTACK (SERVER)

In this section, we motivate and describe the *repeated beam-search attack* which is the major attack algorithm considered in this work. It is a stronger version of the *beam-search attack* described below.

**The Beam-Search Attack (Server)**  Assuming unigram unit, a natural objective of the attacker is to find the sequence $\hat{w}$ with $\hat{w}_t \in \{\tilde{w}_t^1, ..., \tilde{w}_t^N\}$ which is mostly likely to be generated by $P_L$:

$$\arg\max_{\hat{w}} \log P_L(\hat{w}|\tilde{W}_T^N) = \arg\max_{\hat{w}} \sum_{t=1}^T \log P_L(\hat{w}_t|\tilde{W}_{t-1}^N[\hat{w}_{t-1}]). \tag{6}$$

Since the inference for $\hat{w}_t$ only depends on which token is chosen for $\hat{w}_{t-1}$, this optimization problem can be efficiently solved by dynamic programming which maintains the most probable sequence ending with each $w_t^i$ on time-step $t$. The time complexity is $O(N^2T)$. [7] Due to the high similarity with the classical beam-search algorithm, we term it as the *beam-search attack*.

Our experiments (§5) show that the synonym noise scheme §3 is highly vulnerable to the beam-search attack. We show some intuition in the upper part of Figure 2: There does not exist a direct link

---

[6]The average of the true-ratio will always be $\frac{1}{N}$ because each true token is in one of the $N$ hypotheses.

[7]The attacker can reuse saved prediction distributions during generation, and therefore does not need to redo LLM inference. In the bigram case, the time complexity is $O(N^3T)$.

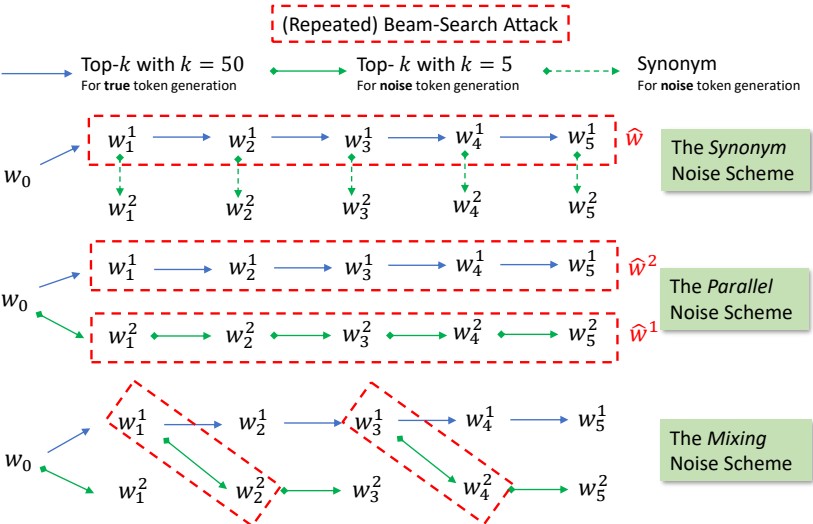

Figure 2: Illustration of different noise schemes under (repeated) beam-search attack. The lattice $W_5^2$ is from the user's perspective and is therefore not permuted. An illustration with a width-3 lattice is given in Figure 5 (§B).

between the noise tokens. The log-probability of the true sequence will likely be higher than any combination of the noise tokens, and is therefore revealed by the attack.

**The Parallel Noise Scheme (User)**   There is an intuitive way to defend against the beam-search attack: The user can sample a noise sequence independent of the true sequence, and make it have higher log-probability than the true sequence by tuning the hyper-parameter of the sampling algorithm. We term it the *parallel noise scheme* and illustrate in the middle of Figure 2.

More concretely, we assume the user is using some popular sampling hyper-parameter for the generation of the true sequence (e.g., $k = 50$ for top-$k$ or $p = 0.96$ for nucleus), which enables the adoption of a more radical hyper-parameter for the sampling of the noise sequences. In our experiments we use $k = 5$. At time-step $t$, the $i$-th ($i > 1$, noise) token is sampled from $P_L(\cdot|\tilde{W}_{t-1}^N[w_{t-1}^i])$. In this way, the noise sequences $w^i$ are parallel and independent of the true sequence $w^1$.

Our experiments show that the parallel noise sequences can very effectively hide the true sequence from the beam-search attack. This motivates our proposed repeated beam-search attack.

**The Repeated Beam-Search (RBS) Attack (Server)**   We propose a simple but more powerful attack algorithm based on the beam-search attack: Given a $N$-lattice, we do beam-search $N - 1$ times. After obtaining the resulting hypothesis sequence of the $i$-th beam-search (denoted as $\hat{w}^i$), we remove the tokens in $\hat{w}^i$ from the lattice, resulting in a $(N - i)$-lattice. After the $(N - 1)$-th beam-search, only one sequence is left in the lattice, which becomes the $N$-th hypothesis $\hat{w}^N$. We term it the repeated beam-search (RBS) attack.

The intuition of why the RBS attack is effective against the parallel noise scheme is shown in the middle of Figure 2. Since the noise sequences are of high probability and independent of each other, it is likely that the $N - 1$ times of beam-search would obtain all the noise sequences as hypotheses which are removed from the lattice in turn, and the remaining true sequence is therefore revealed in the end as $\hat{w}^N$. This would result in a high max-true-ratio.

## 4.2   THE MIXING NOISE SCHEME DEFENSE (USER)

We propose the *mixing noise scheme* to defend against the RBS attack, with the intuition that the true and noise sequences should somehow be mixed.

This scheme can be regarded as a variant of the parallel noise scheme. Again we adopt a radical hyper-parameter for the sampling of the noise sequences (top-$k$ with $k = 5$). At time-step $t$, with a random ratio determined by a hyper-parameter *mix-ratio*, the $i$-th noise token is sampled from

$P_L(\cdot|\tilde{W}_{t-1}^N[w_{t-1}^1])$, **which is the next-token distribution for the true sequence**. [8] Otherwise we sample from $P_L(\cdot|\tilde{W}_{t-1}^N[w_{t-1}^i])$, same as in the parallel scheme.

We illustrate this at the bottom of Figure 2. In comparison to the parallel scheme, the goal is to make the sequence with the highest log-probability be a mix between the true and noise sequences. And the key is to make the true sequence "branch" out to the noise sequences, which breaks the continuity of the noise sequences. Although broken, the radical sampling used for the noise sequence would still attract the repeated beam-search attack, and the true and noise sequences are mixed by the branching connections. Our experiments show that with a tuned mix-ratio, the mixing noise scheme achieves the best max-true-ratio under RBS attack.

## 5 EXPERIMENTS

### 5.1 EXPERIMENT SETTING

**Model & Noise Schemes**   We use the OPT-1.3B (Zhang et al., 2022) model as our base LLM, from which both $P_L$ and $P_M$ are finetuned. In our implementation, for convenience we simulate the user–server interaction protocols (§3.1) on a single machine.

For sampling of the true sequence, we (user) use top-$k$ (Fan et al., 2017) sampling with $k = 50$. For the parallel or mixing noise scheme, $k = 5$ is used. It should be clear that LatticeGen can also be applied to other sampling algorithms with proper hyper-parameters. We limit the maximum generation length to 60 tokens. For the mixing noise scheme, we use a mix-ratio of 0.12 for $N = 2$, and 0.1 for $N = 3$, for the generation part. For the prompt part, we use a mix-ratio of 0.7. They are found to achieve the lowest max-true-ratio on our dev set.

**Dataset & Lattice Finetuning**   Since the word history is noised (discussed in §3.2), LatticeGen is not recommended for tasks with high requirements for consistency or factuality (Pagnoni et al., 2021). In this work we focus on the task of creative writing (Martin et al., 2017), and utilize the WritingPrompts dataset (Fan et al., 2017).

The dataset is composed of stories and the corresponding high-level descriptions as prompts. The average length of prompts/stories is 29/674. We use 200/500 samples from the valid/test set for development/evaluation. The training set (10,000 samples) is used for finetuning of $P_L$ and $P_M$, and we defer details to §A.

**Metrics**   We use a larger LLM, OPT-2.7B, to measure the generation's quality or alignment with the prompt. For quality, we use the popular perplexity metric. For alignment, we use pointwise mutual information (PMI) (Shi et al., 2023):

$$\text{PMI}_{\text{OPT-2.7B}}(x; y) = \frac{\log P_{\text{OPT-2.7B}}(x|y) - \log P_{\text{OPT-2.7B}}(x)}{\text{len}(x)}, \tag{7}$$

where $x$ and $y$ denote the generation and prompt.

To compare between different noise schemes and measure the (semantic) overlap between the attack hypothesis ($\hat{w}$) and the true sequence ($w^1$) under RBS attack, we use the true-ratio or BERTScore discussed in §4. We will report true-ratio for the BS attack and max-true-ratio under RBS attack, and the same applies to BERTScore.

### 5.2 EXPERIMENT RESULTS

Table 1 includes the main results when LatticeGen (LG) is applied to both generation prompt. The standard vanilla model ($P_M$) enjoys the best generation quality (PPL and PMI), while having zero obfuscation (100% true-ratio).

LatticeGen sacrifices generation quality (due to noised history) for obfuscation. The empirical behavior of the three noise schemes aligns with the respective intuition given in §4: The synonym scheme has relatively better PPL&PMI, but is completely defenseless against the BS attack; The parallel scheme is most effective under BS with true-ratio lower than 20%, but is vulnerable under the stronger RBS attack.

---

[8]We will re-sample if the sampled token is the same as the true token.

| Config | $N = 2$ (LG only) | | | | | | $N = 3$ (LG only) | | | | | |
|---|---|---|---|---|---|---|---|---|---|---|---|---|
| Metric / Attack | PPL | PMI | True-Ratio BS | RBS | BERTScore BS | RBS | PPL | PMI | True-Ratio BS | RBS | BERTScore BS | RBS |
| Vanilla ($P_M$), w.o. noise | 28.378 | .340 | 1.0 | 1.0 | 1.0 | 1.0 | / | / | / | / | / | / |
| Synonym, w.o. lattice | 229.616 | .058 | / | / | / | / | / | / | / | / | / | / |
| LG, unigram, synonym | 33.167 | .279 | .923 | .923 | .824 | .824 | 38.267 | .279 | .886 | .886 | .756 | .756 |
| LG, unigram, parallel | 80.071 | .160 | .121 | .878 | .129 | .821 | 105.72 | .141 | .146 | .555 | .133 | .409 |
| LG, unigram, mixing | 73.330 | .190 | .549 | .590 | .389 | .411 | 109.536 | .154 | .328 | .426 | .199 | .252 |
| LG, bigram, synonym | 42.030 | .288 | .987 | .987 | .974 | .974 | 38.005 | .291 | .975 | .975 | .953 | .953 |
| LG, bigram, parallel | 63.124 | .197 | .138 | .861 | .164 | .808 | 71.074 | .144 | .108 | .645 | .141 | .550 |
| LG, bigram, mixing | 64.480 | .232 | .536 | .601 | .409 | .449 | 72.746 | .149 | .383 | .457 | .280 | .318 |

Table 1: Main results when LatticeGen (LG) is applied to both the generation and the prompt. All metrics are the lower the better except PMI. While the generation quality and alignment are degraded, LatticeGen with the proposed mixing scheme successfully protects the true generation from RBS attack to a remarkable degree (measured by max-true-ratio/BERTScore).

The mixing scheme, which is our main recommended scheme, achieves the best protection under the RBS attack. The max-true-ratio/BERTScore is close to or lower than 50%, implying more than 50% of the semantic is hidden from the attacker. There, however, is still a gap between the theoretical best max-true-ratio ($\frac{1}{N}$). The protection is better with $N = 3$, but with worse generation quality.

The quality degradation (especially PPL) is alleviated to some degree by using the bigram units. One could also try trigram or even 4-gram units for further quality improvement. However, the computational cost would grow exponentially and we leave it to future work due to limited resources.

What if we directly apply noise to generation but *without the lattice structure*? We add an additional non-lattice baseline with the same synonym scheme used in LatticeGen: On every time-step, the user gets next-token distribution from the server and generates a true token, but sends a synonym of it back to the server. The finetuning is modified accordingly with details given in §B.4. As shown in Table 1, the synonym noise without lattice results in drastically degraded PPL and PMI. In comparison, LatticeGen provides a trade-off between quality degradation and privacy protection.

Table 2 (§C) compares generation speed of different systems. On the single V100 GPU we use, LG with bigram ($N = 2$) units has a 2x slowdown comparing to $P_M$. Since inference with transformer model benefits from parallel computing, the slowdown should be less significant on servers with stronger computing power.

We show a generation example with RBS attack outputs in Figure 3. LG is able to generate a sample with decent quality. More importantly, much of the story semantics remains hidden from the RBS attack by the mixing noise scheme. More examples and analysis are deferred to §C.

## 6   LIMITATIONS AND FUTURE WORK

LatticeGen sacrifices generation quality and speed for obfuscation of generated contents. While we show the quality degradation can be alleviated to some degree by using larger $m$-gram unit, it would also cause the inference computation to grow exponentially. An interesting future direction is that, instead of running an inference for all $N^m$ grams, we only select a small portion strategically.

In this work we focus on protecting the user and the (repeated) beam-search attack from server. There could be other forms of interesting or stronger attacks on the server side (e.g., manual inspection from a human). On the other hand, sharing generation control with user could also endanger the server (e.g., jailbreaking) (Liu et al., 2023; Li et al., 2023a). We leave further exploration to future work. See §D for more discussions on limitations.

## 7   RELATED WORK

**Lattice in NLP**   The lattice structure has found interesting applications in neural NLP models. As a pioneering work, Su et al. (2017) proposes lattice-based RNN encoders for machine translation, where the lattice is generated by merging results from different segmenters. Buckman & Neubig (2018)

---

**Prompt:** Prompt: You live in a world where light helps you retain and regain memory while darkness makes you forget everything. One day.... Story:

**Generated Text ($P_M$):** I had forgotten everything. The moment when the light shone out of the darkness that my brain had created was when it all came together.Everything. The moment when everything came together, that was when my forgetting started. A slow burn, a warm fire, everything coming back to me. It had been...

**Generated Text (LG):** The world is a strange one, I call it's just that, a big empty, like a dream. The thing I recall was the people. I remember them, but the way they looked and walked, yet 'just a dream. The memory lapse might be about a light, so bright...

**First Round RBS:** *Prompt: You live in a world where light* comes people in memories. It is *darkness, you forget everything. One day.... Story: The world is a strange one, I call it's just* a place I came from. It 'empty'I thought *I recall was the people. I remember them.* I remember them, not as if they were real. '.'*The memory* I most remember is of the people, the...

**Second Round RBS:** applying </ Shogun A are on an underground. the *helps you retain and regain memory while* down *makes* and afraid, until You stumble upon You'unstoppable XIII/r/iN. The surface world I live in is *that, a big empty, like a dream. The thing* as remember most about the same people, over, *but the way they looked and walked, yet 'just a dream.* I think lapse *might be about a light, so bright...*

---

Figure 3: An example of text generation with LatticeGen, using the configuration of bigram, $N$=2 and the mixing scheme. The true tokens are italicized in both rounds of RBS, and the underline indicates that the noise token is mixed from the previous true token. Note that the prompt is also noised by LG. See §C for more examples.

proposes a neural lattice language model, which constructs a lattice of possible paths (segmentations) through a sentence in order to model multiple granularities. Lattice-BERT (Lai et al., 2021) trains LLM to predict a masked portion of a lattice representing possible segmentations of a sentence. To the best of our knowledge, our work is the first to utilize the lattice structure for privacy-aware generation.

**Differential Privacy (DP) for LM Training and Inference**   There are numerous existing works on how to train LLMs with differential privacy (Li et al., 2021; Yu et al., 2021; Kerrigan et al., 2020), which mostly rely on DP-SGD (Abadi et al., 2016) and limits leakage of private data during training. More related to LatticeGen is a line of work with local DP (Xu et al., 2020; Meehan et al., 2022), which applies discrete noise onto text and can be used to synthesize private text data (Yue et al., 2023; Mireshghallah et al., 2023). Finally, there has been recent work on protecting data for in-context learning (Panda et al., 2023; Duan et al., 2023) or prefix tuning (Li et al., 2023b).

It is not directly clear how these techniques can be adapted for our setting of privacy-aware autoregressive text generation. In comparison, LatticeGen provides a totally different and cooperative approach with the lattice structure and novel defense and attack schemes. Due to lack of space, we defer discussion of **Homomorphic Encryption** and **Prompt Anonymization** to §E.

## 8  CONCLUSION

LatticeGen aims for an ambitious and seemingly conflicting goal: The server still does most computation for the generation but does not know what exactly is generated. This is achieved by our proposed noised lattice structure, and a cooperative generation protocol between the server and user.

While the noised lattice degrades generation quality and inference speed, LatticeGen with our proposed mixing noise scheme successfully prevents a malicious server from recovering the true generation to a remarkable degree (more than 50% of the semantic remains unknown as measured by BERTScore). We hope our work could inspire more research into this under-studied yet important field of privacy-aware LLM generation on cloud.

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

SUPPLEMENTAL MATERIALS

## A   MODEL TRAINING AND INFERENCE WITH LATTICE (SERVER)

**Finetuning with Linearized Lattice Format**   We now describe how $P_L$ is obtained by finetuning a standard autoregressive LM $P_M$ parameterized by $\theta$ to accept and make next-token predictions on a linearized $N$-lattice. We assume access to a public corpus $D$ for finetuning. For simplicity, we focus on the training objective for one length-$T$ sentence $w^d \in D$ and we also assume $N = 2$ (the process for $N > 2$ is highly similar) and unigram units.

We first generate the lattice $\tilde{W}_T^2$ for $w^d$ with a given noise generation scheme. The tokens in the data $w^d$ will be used as the true tokens $w_t^1 := w_t^d$ and we need to generate the noise tokens $w_t^2 \neq w_t^1$ for each time-step $t$. In our experiments we use a simple synonym scheme (§3.1).

The noise generation scheme used by server in the finetuning stage might be different from the scheme used by user in the actual generation, but empirically we find this misalignment does not affect the generation performance drastically. To be consistent with the actual generation protocols, the tokens on each time-step are shuffled and the positions of the true tokens need to be saved.

The goal is to finetune the LLM for next-token prediction for tokens in the linearized lattice. The challenge is that we do not have ground-truth next token for the noise tokens $w_t^2$. Instead of generating pseudo training data, we utilize the property that the lattice is shuffled on each time-step, and simply omit the labels (no training signal) for the noise tokens. The intuition is that since the token positions are randomly shuffled, after training the LLM will be able to predict the next token for *any* position in the lineazried lattice and we find this simple finetuning strategy works well in practice.

In summary, we only train the LLM for predict the next token for the true tokens $w_t^1 = w_t^d$ in $\tilde{W}_T^2$ (illustrated in Figure 4). We summarize it into the following objective:

$$\mathcal{L}_{\text{lattice-finetune}}(w^d, \tilde{W}_T^2; \theta) = \frac{1}{T} \sum_{t=1}^{T} \log P_\theta(w_t^1 | \tilde{W}_{t-1}^2[w_{t-1}^1]). \tag{8}$$

The implementation is similar to the standard finetuning of autoregressive LMs, and we only need to make modifications to the inputs and the labels.

Figure 4: An illustration of the lattice-finetuning objective described in §A. The input is a linearized 2-lattice permutated on each time-step. The noise tokens $w_t^2$ do not get training signal.

**Inference**   We now discuss how the server can do efficient LLM inference at time-step $t$. Since linearize$(\tilde{W}_{t-2}^N)$ from the previous time-step $t-2$ is a prefix of linearize$(\tilde{W}_{t-1}^N)$, the server can reuse the saved LLM hidden states[9] from the last time-step for the inference of $\{P_L(\cdot | \tilde{W}_{t-1}^N[\tilde{w}_{t-1}^i])\}_{i=1}^{N}$. In this way, none of the computations on the server-side are repeated and the computation cost remains reasonable.

**Implementation Details**   Our model implementation, training and inference utilize the HuggingFace transformers library (Wolf et al., 2020). We finetune $P_L$ with learning rate of $10^{-4}$ and a batch size of 8 for 3 epochs using the PyTorch (Paszke et al., 2019) implementation of the AdamW (Loshchilov & Hutter, 2017) optimizer. We perform finetuning of the model under various configurations on one Nvidia V100 GPU.

---

[9]The `past_key_values` in HuggingFace transformers library.

---

**Algorithm 1** Pseudo-code for LatticeGen (Unigram)

---

**Input (Server):** Lattice-finetuned LLM $P_L$, lattice width $N$, generation length $T$.
**Input (User):** Prompt $p$, a noise generation scheme $S$, a private large prime number for random seed.
 1: User sets $\tilde{w}_0^i := $ `<bos>` for $1 \leq i \leq N$. And initialize the reverse permutation as the identity mapping.
 2: The server begins with an empty lattice.
 3: The user sends $[\tilde{w}_0^1, ..., \tilde{w}_0^N]$ to server indicating the beginning of generation.
 4: **for** $t = 1 \ldots T$ **do**
 5:     # Server Steps Below
 6:     Receives $[\tilde{w}_{t-1}^1, ..., \tilde{w}_{t-1}^N]$ from user and use it to extend the lattice to $\tilde{W}_{t-1}^N$.
 7:     Infer the LLM $P_L$ and obtain $\{P_L(\cdot|\tilde{W}_{t-1}^N[\tilde{w}_{t-1}^i])\}_{i=1}^N$.
 8:     The next-token distributions are sent to the user as $N$ length-$|V|$ vectors.
 9:     # User Steps Below
10:     Receives the next-token distributions $\{P_L(\cdot|\tilde{W}_{t-1}^N[\tilde{w}_{t-1}^i])\}_{i=1}^N$ from server.
11:     Apply the saved reverse permutation mapping to get $\{P_L(\cdot|\tilde{W}_{t-1}^N[w_{t-1}^i])\}_{i=1}^N$.
12:     **if** $t \leq \text{len}(p)$ **then**
13:         Set $w_t^1 := p_t$.
14:     **else**
15:         Sample $w_t^1$ from $P_L(\cdot|\tilde{W}_{t-1}^N[w_{t-1}^1])$.
16:     **end if**
17:     Generate $N - 1$ noise tokens $\{w_t^2, ..., w_t^N\}$ with scheme $S$.
18:     Set the current random seed to be $t$ multiplied by the private prime number.
19:     Obtain the permuted list $[\tilde{w}_t^1, ..., \tilde{w}_t^N]$ from the current private random seed.
20:     Save the reversing permutation for next time-step.
21:     Send $[\tilde{w}_t^1, ..., \tilde{w}_t^N]$ to server.
22: **end for**
**Output (Server):** Permutated lattice $\tilde{W}_T^N$.
**Output (User):** True sequence $w^1$, and lattice $W_T^N$.

---

## B    AUXILIARY FRAMEWORK DESCRIPTION

We summarize the LatticeGen protocols as pseudo-code in Algorithm 1. [10] The discussion about noise schemes is in §4.

An illustration of various noise schemes with a width-3 lattice is provided in Figure 5.

### B.1    INCORPORATING THE PROMPT (USER)

The prompt $p$ can be easily incorporated by the following. At all time-steps $t$ with $t \leq \text{len}(p)$, instead of sampling $w_t^1$ from $P_L(\cdot|\tilde{W}_{t-1}^N[w_{t-1}^1])$, the user directly sets $w_t^1 := p_t$. All other steps in the protocols including the noise token generation continue as normal. In this way, the prompt is also embedded and noised in the lattice.

### B.2    INCORPORATING BIGRAM UNITS

We explore a variant in which we expand the unit from unigram (one token) to bigram. While **the lattice is still one token per time-step**, the user enumerates all $N^2$ potential bi-gram combinations of $w_{t-2}$ and $w_{t-1}$ and ask the server LLM to return the next-token prediction distribution for each bigram. We illustrate it in Figure 6. Accordingly, the finetuning stage (§A) needs to be modified so that the model treats bigram instead of unigram as the unit.

In this way, the approximate probability of generating a true sequence $w$ is (following §3.2):

$$\log P_{L\text{-bg}}(w) \approx \sum_{t=1}^{T} \log P_{L\text{-bg}}(w_t|\tilde{W}_{t-1}^N[w_{t-2}w_{t-1}]), \tag{9}$$

where $P_{L\text{-bg}}$ can utilize the exact bigram context (to be compared with Equation 2). In experiments, we observe visible improvement in generation quality comparing to the unigram version. However

---

[10]For convenience, in the beginning, the user sends out $N$ `<bos>` tokens to server.

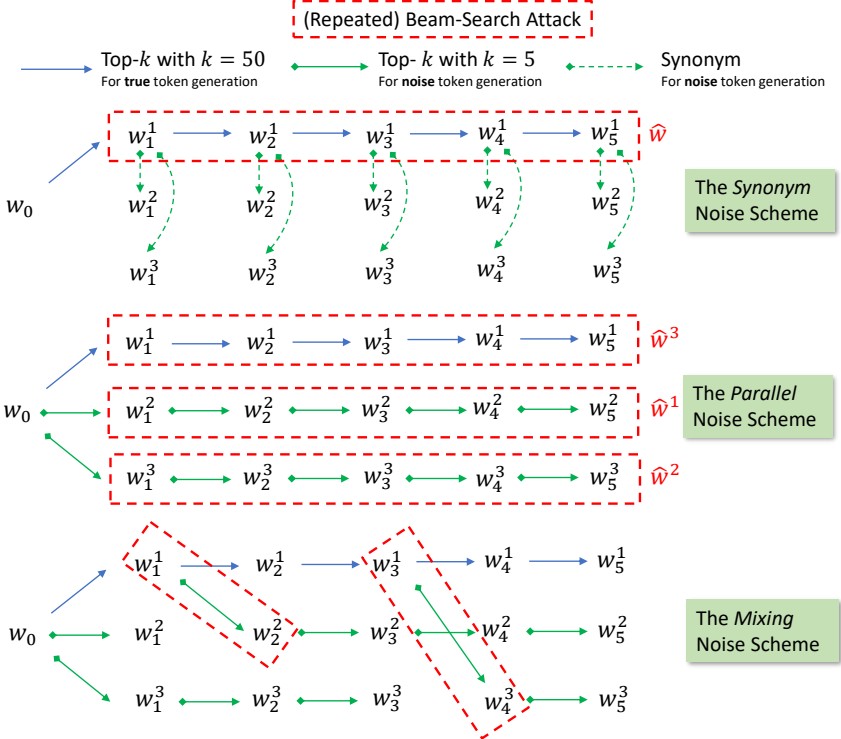

Figure 5: Illustration of different noise schemes under (repeated) beam-search attack. The lattice $W_5^3$ is from the user's perspective and is therefore not permutated.

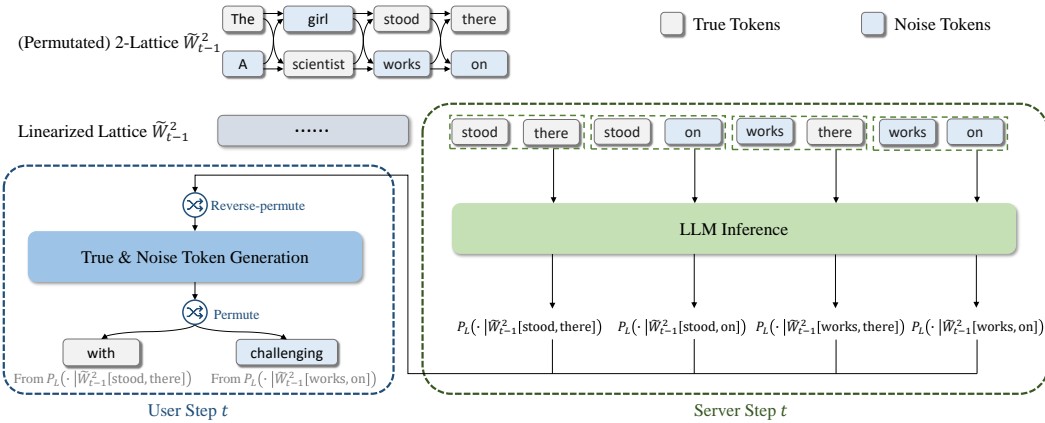

Figure 6: User–Server interaction under LatticeGen with bigram units for time-step $t$.

on each time-step, the server needs to inference the LLM on input of length $2N^2$, instead of length $N$ in the unigram case. **The inference speed is traded for generation quality.**

We end this section by emphasizing that the bigram variant mostly affects LLM inference and does not change the lattice structure. Therefore it does not affect the noise schemes to be discussed in §4.

## B.3 COMMUNICATION COST

At each time-step, the server needs to send user $N$ (or $N^2$ in the bigram case) length-$|V|$ vectors, which could be slow if $|V|$ is large. This can be largely alleviated if the user and server can agree

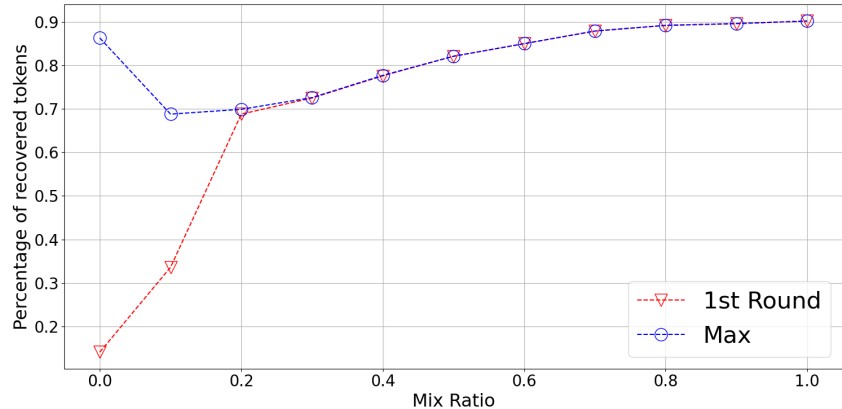

(a) Max-true-ratio under different mix-ratio for N=2.

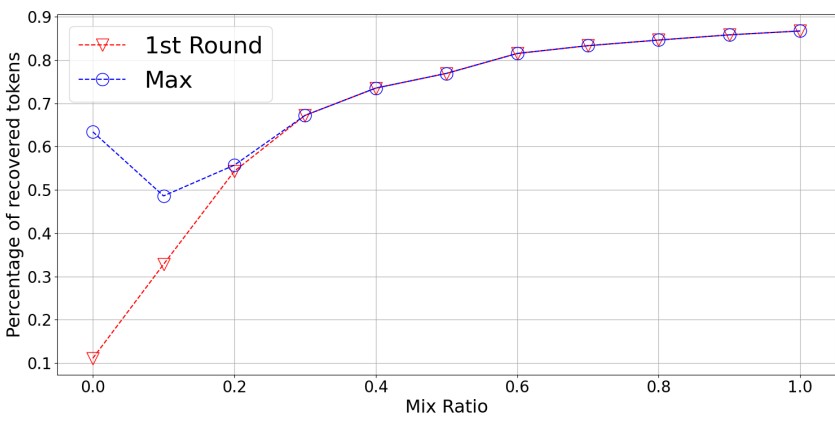

(b) Max-true-ratio under different mix-ratio for N=3.

Figure 7: How tuning of mix-ratio affects the result from RBS attack. Bigram units are used.

upon a sampling algorithm beforehand. For example, if top-$k$ sampling with $k = 50$ is used, then only the logits and indices of the top-50 tokens are needed.

### B.4 THE NON-LATTICE BASELINE

The training for the non-lattice baseline is a bit similar to the lattice finetuning process described in §A, with the difference that the true tokens are not included in the input. Following the notations in §A with $w^d$ as the data sample and $w^2$ as its synonym noise sequence, the training objective is formulated as:

$$\mathcal{L}_{\text{non-lattice,synonym}}(w^d, w^2; \theta) = \frac{1}{T} \sum_{t=1}^{T} \log P_\theta(w_t^d | w_{0..t-1}^2). \tag{10}$$

Basically, the model is trained to predict the next true token with all input tokens noised.

## C AUXILIARY RESULTS

Figure 7 shows the impact of mix-ratio on max-true-ratio under RBS attack. When mix-ratio=0, it is reduced to the parallel scheme and the true-ratio from the 1st BS is very low but the max-true-ratio is high. As mix-ratio increases, more true tokens are mixed in to the 1st beam. The mix-ratio achieveing the best max-true-ratio is around 0.1.

| Speed (second/token) | N=1 | N=2 | N=3 |
|:---:|:---:|:---:|:---:|
| $P_M$ | .061 | / | / |
| LG, Unigram | / | .088 (1.44x) | .125 (2.04x) |
| LG, Bigram | / | .127 (2.08x) | .186 (3.04x) |

Table 2: Generation speed comparison between different systems. For LG, the mixing noise scheme is used. Our implementation is run on a single V100 GPU.

Similar to Figure 3, Figure 8 shows an example using a different prompt using bigram $N = 2$, and Figure 9 shows an example of generation with the same prompt as in Figure 3 using $N = 3$, both examples contain sample lattices of prompt tokens, each separated by a vertical bar.

On a single V100 GPU we use, LG with bigram units ($N = 2$) has a 2x slowdown comparing to $P_M$ (Table 2, §C). Since inference with transformer model benefits from parallel computing, the slowdown should be less significant on servers with stronger computing power.

## D   MORE DISCUSSION ON LIMITATIONS AND FUTURE WORK

This section continues from §6.

In the current implementation, we lattice-finetune a seperate OPT model for every different lattice configuration, which is space unfriendly. As future work, it would be interesting to explore a unified format of linearized lattice by which a single LLM can process different lattice configurations.

## E   MORE DISCUSSION ON RELATED WORK

This section continues from §7.

**Homomorphic Encryption**   There is also a line of work (Hou et al., 2023; Chen et al., 2022) applying techniques from homomorphic encryption (Rivest et al., 1978; Gentry, 2009) to transformer LM. While they enjoy nice cryptographic guarantees, the induced computational cost is usually huge.

**Prompt Anonymization**   Contemporary and independent of our work, Chen et al. (2023) proposes to anonymize the named entities (e.g., change USA to <GPE>) in the prompt, and de-anonymize after receiving the generated text from server. In comparison, LatticeGen offers a more general option in that all types of tokens can be noised.

## F   THE CURRENT PRIVACY PROTECTION PRACTICES IN INDUSTRY

The privacy protection offered by today's LLM providers' data usage and retention policies is far from enough. [11] For example, OpenAI's consumer-facing ChatGPT used to train its models with user input, and also shares user input with third-party providers, and Google's Bard retains user activity for at least 3 months. As a striking example, employees in Samsung reportedly shared sensitive code with OpenAI during their interaction with ChatGPT. [12] More recently, some of the users' conversations with Bard is mistakenly indexed and accessed by Google search. [13]

While providers have recently improved their security posture (e.g., OpenAI no longer uses data submitted via its API to train its model), users still can not assume that all sent/received data will be immediately and completely deleted. Rather than regulations, our proposed LatticeGen takes an algorithmic and cooperative approach to give the user advantage and control in privacy protection.

---

[11] https://opaque.co/announcing-opaqueprompts-hide-your-sensitive-data-from-llms/

[12] https://gizmodo.com/chatgpt-ai-samsung-employees-leak-data-1850307376

[13] https://venturebeat.com/ai/oops-google-search-caught-publicly-indexing-users-conversations-with-bard-ai/

**Prompt:** Prompt: It is believed that the cure for cancer has been discovered on Mars, astronauts have only one chance to bring it home where they hope to duplicate its properties for mass distribution. Story:

**Generated Text ($P_M$):** The quick spreading of disease was what excited the scientist most about the data he had brought the team back from the brink of extinction. A species capable of living for hundreds of thousands of years using a system of biological engineering had finally proved the impossible, he was excited. Cancer could be cured. It was

**Sample Linearized Lattice of Prompt:** ... | atom Write: Write: It atom It | Write'It is It'Write is | is the'the is believed'believed | believed that the belief believed belief the that | ...

**Generated Text (LG):** " You've got to save... my crew shouted. We had done everything possible in the best medical attention. We trained our astronauts as best science. " Yeah, the best in the world could even the journalists asked the President can ever be brought back. " asked " No.

**First Round RBS:** *Prompt: It is believed* belief of a *for cancer has been* lost over the last *astronauts* are on *one* month *to* get *it home* planet to *hope to* save *its* people. Story:. *Story: "* The last chance to make it. " It was a group of seven. We left the planet available to us in the best. We had the best medical team on *the best in the* press said. " The best *can ever be brought back. " " No.*

**Second Round RBS:** Ukrainiansb atom Write'the that the cure is a that will *discovered on Mars,* and *have only* been *chance* left *bring* the cure *where they* can it *duplicate. properties for mass distribution* "Prompt: *You've got to save... my crew shouted. We had done everything possible in the best medical attention. We trained our astronauts as best science. " Yeah,* Mars. " a *world could even the journalists asked the President* cancer cure in Mars? " We *asked* " Why

Figure 8: Another example of text generation with LatticeGen, using the configuration of bigram, $N$=2 and the the mixing scheme. The true tokens are italicized in both rounds of RBS, and the underline indicates that the noise token is mixed from the previous true token. Note that the prompt is also noised by LG.

---

**Prompt:** Prompt: You live in a world where light helps you retain and regain memory while darkness makes you forget everything. One day.... Story:

**Sample Linearized Lattice of Prompt:** ... | You are You story The are A live You live A are A story The live The story | areasia live under liveasia story in live in are in are under story under storyasia | in a under Madagascar under aasia the in the in Madagascar under theasia aasia Madagascar | the world a city the, Madagascar world Madagascar, the city Madagascar city a world a, | ...

**Generated Text (LG):** I had become thin. They could barely visible in the further than before. The buildings that surround me like a surround me. I could feel my brain cells lining the walls and outside me, as my brain putting the whole society would be it. I would never get used to the outside world, my

**First Round RBS:** *Prompt: You live in a world where* people are people, *and* can consciousness *while* sleeping and dreaming *forget*ful. *One day,*you.. *Story:* The air was thick with the city far, far more clearly *than before. The buildings* and emotions, *like a surround me. I could feel my brain cells lining the* inside me, as if I was surrounded by so many thoughts, not just. *I* could feel my body, or at least. My

**Second Round RBS:** guilt: A The story under the, in order *helps you retain* consciousness- *memory, darkness makes you* see *everything* that The night.. You do nically trained my eyes. *They could barely visible* from my vision, as I felt my mind had become one would in a shell. I could see the thoughts firing up in a massive wall. It had been this way of thinking and acting. I *never get used to* this coldness*, my*

**Third Round RBS:** ief :990 A are asia Madagascar city that *light* to us and can *regain* their of asleep surrounds sure a. from You man your You go Finch POLIT *I had become thin* to see what was in *in the further* then my surroundings. Darkness *that surround me* I was hurricane around me- It had become a starting becoming more *walls and outside me, as my brain putting the whole society would be it* one way would could feel my brain *the outside world* that again

Figure 9: An example of text generation with LatticeGen, using the configuration of bigram, $N$=3 and the the mixing scheme. The true tokens are italicized in all rounds of RBS, and the underline indicates that the noise token is mixed from the previous true token. Note that the prompt is also noised by LG.

