# OpenReview forum: "LatticeGen: A Cooperative Framework Which Hides Generated Text in A Lattice For Privacy-Aware Generation on Cloud"
_ICLR.cc/2024/Conference — ICLR 2024 Conference Withdrawn Submission_

### Official Review · Reviewer_hyc5 · 2023-10-30

**Soundness:** 3 good
**Presentation:** 3 good
**Contribution:** 2 fair
**Rating:** 5
**Confidence:** 3

**Summary:**

The paper introduces a framework named LatticeGen, which protects the privacy of user text and generated text through a method of generating token lattices via interaction between the user end and the server. Additionally, the authors propose a potential attack method, the beam-search attack, and introduce an analytical metric. They also analyze how using LatticeGen can protect privacy.

**Strengths:**

1.Compared to traditional NLP encryption methods, this paper considers the approach of simultaneously obfuscating both the user-uploaded prompts and the generated text, which aligns well with the inherent privacy requirements of Large Language Models (LLMs).

2.The paper also addresses potential vulnerabilities by proposing a potential attack method, the beam-search attack, and subsequently introduces a metric for analyzing privacy protection under this attack.

Besides, the clarity of the paper is commendable, with logical flow, well-defined terms, and illustrative examples ensuring accessibility to a broad audience. The use of figures enhances understanding, making complex concepts more digestible.

**Weaknesses:**

1.From a motivational perspective, the necessity of a collaborative and interactive process between the client-side and the server for text generation is questionable. Often, users utilizing large language models are primarily interested in obtaining results, rather than performing operations on the data. Moreover, they may not necessarily possess the computational power required for such operations.

2.The authors utilize the metric max-true-ratio to demonstrate how their privacy-preserving mechanism can withstand attacks. However, it is crucial to acknowledge that a sentence inherently possesses its own structure and semantics, and often, obfuscating just a part of it may not suffice to protect the privacy of the entire sentence. The authors do mention that there is often a trade-off between privacy and model effectiveness. In scenarios where not enough words in a sentence are obscured, it raises a concern whether the semantics of the sentence could still be exposed. This aspect deserves further attention to ensure comprehensive privacy protection.

**Questions:**

1.Under the attack scenario presented in this paper, could you elucidate why the collaborative and interactive text generation process is a necessary operation to protect privacy?

2.Does a max-true-ratio of 50% ensure that the semantics of the original text are protected from being disclosed?

3.Can the framework presented in this paper withstand other common attacks targeted at Large Language Models (LLMs), apart from the attack method proposed in the article?

---

> ### Author Response · Authors · 2023-11-16
> **Thanks for the review! Please read our response.**
>
> Thanks for the review! Here are our response to your concerns.
>
> > From a motivational perspective, the necessity of a collaborative and interactive process between the client-side and the server for text generation is questionable. Often, users utilizing large language models are primarily interested in obtaining results, rather than performing operations on the data. Moreover, they may not necessarily possess the computational power required for such operations.
>
> We motivate privacy-aware generation in the beginning of Section 2, which we repeat here: A key reason is that in many scenarios, the generation from the LLM can affect the user’s private real-life decisions: e.g., a customer is likely to go to the restaurant suggested by the LLM; an engineer could adopt the code proposed by the LLM; a writer could take inspiration from outputs provided by the LLM. In all those cases, the user would prefer that the generated text can be kept private. Also see Appendix F for recent privacy-related incidents with ChatGPT or Bard. The obfuscation provided by LatticeGen makes it harder for a hypothetically malicious server to infer the user’s actions after interacting with the LLM.
>
> Next, the computational demands on the client side are really minimal, focusing primarily on sampling operations and token permutations, which should be easy for any laptop. The protocol on the client (user) side can be implemented as a simple python script, and does not require human labor.
>
> > Does a max-true-ratio of 50% ensure that the semantics of the original text are protected from being disclosed?
>
> The max-true-ratio only considers exact match. Therefore, we also utilize BERTScore [1] to measure the leaked semantic under attack. It is introduced in the beginning of Section 4. As shown in Table 1, the leaked semantic are around 40% when N=2 and around 30% when N=3, which is remarkable since the current user-server interaction paradigm provides zero protection (100% leakage).
>
> [1] BERTSCORE: EVALUATING TEXT GENERATION WITH BERT. ICLR 2020
>
> > Can the framework presented in this paper withstand other common attacks targeted at Large Language Models (LLMs), apart from the attack method proposed in the article?
>
> While we agree that the attacks we consider are not exhaustive, we want to emphasize that we are the first to introduce a client-server interaction for privacy-aware generation. Therefore, our proposed attack strategy (repeated beam-search attack) serves as a poineering evaluation method in the absence of established benchmarks. In our experiments, it is already shown to be very effective against the synonym and parallel defense schemes. In Section 6 on Limitations, we explicitly recognize that there could be other attack approaches, which we leave for future research to explore. If the reviewer has any particular attack in mind, we would be happy to try.
>
> We would be happy to address any addition question or concern, thanks!

---

### Official Review · Reviewer_RjUf · 2023-10-31

**Soundness:** 2 fair
**Presentation:** 3 good
**Contribution:** 2 fair
**Rating:** 6
**Confidence:** 2

**Summary:**

This paper considers the setting of a server computing the result of running a language model on user input and suggests the following approach to hiding the generated text (and the user prompt) from the server. instead of being given the prompt tokens by the user the user instead gives a collection on N tokens in a lattice which the server can then (at little extra cost) run the model on all of at once (up to some approximation) the user is then given N possibilities for the next token and knows which one is correct because it set the lattice up. The paper analyses a couple of rounds of attack/defence against this protocol and presents an experimental analysis of the semantic leakage and the accuracy.

**Strengths:**

The paper does a reasonably thorough job of considering attacks, granted they are not aiming to be exhaustive as would be very hard for something like this.
The use of lattice techniques for this purpose seems like an interesting idea to me (though I am not an expert by any means) and thus something that was worth exploring.

**Weaknesses:**

The intersection of the degradation in quality and the fact that by their own measure the model leaks about 40% of the semantic content anyway makes this not a clearly useful idea.
Whilst I don't know if it would be feasible in the space available tht fact that I can read the paper and not have much idea what a lattice is afterwards does seem unsatisfying as it seems to be the main technique being brought to bear.

**Questions:**

Could the idea of what is going on with this lattice be made clearer here?
Can you motivate the importance of hiding generated output?

---

> ### Author Response · Authors · 2023-11-16
> **Thanks for the review! Please read our response.**
>
> Thanks for the review! Here are our response to your concerns.
>
> >The intersection of the degradation in quality and the fact that by their own measure the model leaks about 40% of the semantic content anyway makes this not a clearly useful idea.
>
> As stated in the beginning of Section 4, under repeated beam-search attack and the lattice structure, 1/N is a lower bound for max-true-ratio. As shown in Table 1, our proposed mixing scheme has effectively come close to this bound. Users can use a larger N for more protection. Also please note that our work represents a pioneering effort in privacy-aware generation. The transition from 100% (zero protection) to around 40% leakage marks a significant step.
>
> > Could the idea of what is going on with this lattice be made clearer here?
>
> We would appreciate if the reviewer can ask a more specific question. Our generation protocol is given in Section 3.1 (Algorithm 1), and illustrated in Figure 1. The key idea is that on each time-step, in addition to the true token, the user also generates N-1 noise tokens. The server does not know which one is the true one, and gives next-token prediction for all tokens.
>
> > Can you motivate the importance of hiding generated text?
>
> In the beginning of Section 2, we discuss our motivations for privacy-aware generation, which we repeat here: A key reason is that in many scenarios, the generation from the LLM can affect the user’s private real-life decisions: e.g., a customer is likely to go to the restaurant suggested by the LLM; an engineer could adopt the code proposed by the LLM; a writer could take inspiration from outputs provided by the LLM. In all those cases, the user would prefer that the generated text can be kept private. Also see Appendix F for recent privacy-related incidents with ChatGPT or Bard. The obfuscation provided by LatticeGen makes it harder for a hypothetically malicious server to infer the user’s actions after interacting with the LLM.
>
> We would be happy to address any addition question or concern, thanks!

---

### Official Review · Reviewer_s2Jy · 2023-11-03

**Soundness:** 2 fair
**Presentation:** 1 poor
**Contribution:** 2 fair
**Rating:** 3
**Confidence:** 2

**Summary:**

This paper studies how to let a server generate text using a large language model such that the generated text received by the client is private.

**Strengths:**

The problem of keeping generated text private is an interesting one.  The paper contains ideas of how certain elements of an algorithm for this task could be performed.

**Weaknesses:**

The paper (and the preliminaries section) don't introduce all terminology and concepts.  While a reader may potentially read all cited work and try to understand in this way the text, the text is insufficiently self-contained to be easily digestible by a non-expert.  For example,
* The preliminaries section says that a lattice structure is used and cites two papers (which explain different things) but doesn't provide a clean definition of "lattice".  As probably "lattice" in not meant in the purely mathematical sense (a partial order with least upper bound and greatest lower bound operators), more details are needed.
* The paper doesn't introduce "transformer".
* In "As the name suggests, we conduct a simple linearization operation before feeding it to the LM" it is unclear to what "it" refers, and it is unclear on what object the "linearization" operation is performed.
* In Sec 3.1, there is a "lattice-finetuned LLM", but the text doesn't explain what this means.


Also at other points, the presentation is hard to follow.  For example, at the bottom of page 3 the text starts to describe an algorithm informally, but there is no pseudo-code or other algorithm formalization which may help the reader to get a more precise view of what is intended.


The text makes no precise formal claims, but mainly describes an approach.
The paper presents a number of experiments, essentially investigating how robust the proposed approach is to a set of attacks the authors have selected (without much motivation on why defending against these attacks is sufficient).


There are quite a few language issues, e.g., missing articles before "cloud" (the cloud, a cloud, ...)

**Questions:**

--

**Details Of Ethics Concerns:**

--

---

> ### Author Response · Authors · 2023-11-16
> **Thanks for the review. Please read our response.**
>
> Thanks for the review. We want to emphasize that many definitions and details which claimed to be missing by the reviewer are in fact included in our manuscript. We detail them below. We sincerely hope that based on our response, the reviewer can read our paper again and re-assess our work.
>
> > ...doesn't provide a clean definition of "lattice"...
>
> It is defined in Section 2 and illustrated in Figure 1. It's quite simple. For a width-N lattice, each time-step contains N token options.
>
> > The paper doesn't introduce "transformer".
>
> We have cited the "Attention is all you need." paper in the first appearance of "transformer". The transformer model is ubiquitous in NLP and machine learning papers and usually does not need introduction. Moreover, readers do not need to understand transformer for LatticeGen, the only key background needed is autoregressive LM generation and is detailed (in bold) in Section 2.
>
> > In "As the name suggests, we conduct a simple linearization operation before feeding it to the LM" it is unclear to what "it" refers, and it is unclear on what object the "linearization" operation is performed.
>
> This sentence continues from the previous paragraph and "it" refers to the lattice (we will change it in the revision to be precise). Since this paragraph is named "Linearized Lattice", it should be clear that the linearization operation is performed on the lattice (also see Equation 1).
>
> > there is no pseudo-code or other algorithm formalization which may help the reader to get a more precise view of what is intended.
>
> In Page 3 and appendix, we provide Figure 1 and Algorithm 1 (pseudo-code, page 15) to help understand our proposed protocol. We will move the pseudo-code to main text in our revision.
>
> > The text makes no precise formal claims, but mainly describes an approach. The paper presents a number of experiments, essentially investigating how robust the proposed approach is to a set of attacks the authors have selected (without much motivation on why defending against these attacks is sufficient).
>
> While we agree that the attacks we consider are not exhaustive, we want to emphasize that we are the first to introduce a client-server interaction for privacy-aware generation. Therefore, our proposed intuitive attack strategy (beam-search attack) serves as a poineering evaluation method in the absence of established benchmarks. In our experiments, it is already shown to be very effective against the synonym and parallel defense schemes. In Section 6 on Limitations, we explicitly recognize that there could be other attack approaches, which we leave for future research to explore. If the reviewer has any particular attack approach in mind, we would be happy to try.
>
> > There are quite a few language issues, e.g., missing articles before "cloud" (the cloud, a cloud, ...)
>
> We will add the missing articles, thanks!
>
> We would be happy to address any addition question or concern, thanks!

---

> > ### Comment · Reviewer_s2Jy · 2023-11-16
> >
> > Unfortunately, despite renew search, I can't find in section 2 a definition for "lattice".
> > In particular, I read in Section 2 the following:
> >
> > > The Lattice .... (Young et al., 2006),
> >
> > This is a reference to another paper
> >
> > > which is ... widely used ... to represent a range of hypotheses.
> >
> > This suggests to the reader that "hypothesis" is important, but the word won't re-occur in the text for 3 pages.
> >
> > > ... we adopt a ... form of lattice ... (Mangu et al., 1999).
> >
> > A reference to another paper, with another view on what exactly is a "lattice".
> >
> > > ... example is shown in the left part of Figure 1.
> >
> > Figure 1 doesn't provide a definition, and shows many other things too
> >
> > > For a width-N ...
> >
> > From here on, the authors seems to assume that the reader knows the specific type of lattice they have in mind, including that it contains time steps and has a width (of which we also don't have a definition).
> >
> > In summary, there is a lot of text, but not a *definition* of "lattice".
> > If a diligent reader looks at Wikipedia, he will find, next to 6 general meanings of the word "lattice" and 7 meanings in science, that there are at least 7 meanings only already in the field of mathematics.   None of the 9 wikipedia pages referenced in that section contain the word "width", so it seems the authors use here a non-mainstream notion
> >
> > Hence, the claim that the term is defined in Section 2 seems false, while a definition is appropriate given the non-standard meaning.
> >
> > >>  ... it is unclear to what "it" refers, and it is unclear on what object the "linearization" operation is performed.
> >
> > > This sentence continues from the previous paragraph and "it" refers to the lattice (we will change it in the revision to be precise). Since this paragraph is named "Linearized Lattice", it should be clear that the linearization operation is performed on the lattice (also see Equation 1).
> >
> > The previous paragraph ends with the sentence "Below we first introduce this format, and describe the finetuning objective."  This previous sentence doesn't seem to contain something which you can linearize.  In particular, it seems you don't linearize a finetuning objective, and "this format" already refers to (in the previous sentence) "linearized lattice format".  It seems strange to linearize a "linearized lattice format" since by its name it has already been linearized.  So in the end, the first candidate we find which may match "it" seems to be the "base LLM $P_M$", which according to your answer above it not what "it" was referring to.
> >
> > There seems also to be some unexplained distinction between a "lattice" and a "lattice format", since the paragraph mentions both with the adjective "linearized".  It is hence not obvious that "it" was referring to the lattice and not to the format.  In fact, the title of the paragraph contains "linearized lattice format" and not "linearized lattice" as you claim.
> >
> > Equation 1 does not bring much clarity.  It contains w_i and <bos>, which are tokens according to the paragraph "Standard Autoregressive LM Generation" (even if that paragraph contains but doesn't define $w_{0...t-1}$ which I assume to be $w_0 ... w_{t-1}$) and contains ${\tilde{W}}_N^T$.  The latter,  ${\tilde{W}}_N^T$, seems to be a lattice on which some shuffle operation has been performed.  The name "token options" at some point did me believe that a lattice allows to select in any time step any of the several "token options" available for that time step, but I discarded that possible interpretation as the sentence "We will also use the notation ${\tilde{W}}_N^t$  to emphasize that the tokens in the lattice are shuffled in each time-step." suggests that the token options (or tokens?) in every time steps of the lattice have a particular order (which wouldn't be the case if they would just be parallel options).  It is of course possible the shuffling is intended to hide information, but in that case it is unclear why shuffling only "inside a time step" (rather than shuffling also tokens from different time steps) already hides sufficiently.  Of course, all of this is guessing since Equation 1 uses ${\tilde{W}}_T^N$ rather than ${\tilde{W}}_t^N$ without explaining whether the difference between $T$ and $t$ is significant.  On the right hand side of Eq 1 we see two operations, "+" and "concat".  It is unclear what is the difference between both operations.   Anyway, the equation does not clarify what is formally the structure of a lattice, nor how the linearization works (which, after all, should not be so difficult to formalize precisely).
> >
> > Page 15 indeed contains pseudo-code vaguely describing LatticeGen in natural language, but without much mathematical precision.
> >
> > My domain is not NLP, I only know about grammars, automata, DNN, privacy and a few other topics.  Possibly other readers can understand it easily.  The above only illustrates that a reader needs to do a lot of guessing / inference / disambiguation, which is not the case in well written papers.

---

> > > ### Author Response · Authors · 2023-11-16
> > > **Thanks for the response.**
> > >
> > > We sincerely thank you for the response. It is very helpful to get comments from non-NLP researchers. We will improve our writing for non-NLP researchers in our revision.